# Hierarchies and Dominance Behaviors in European Pond Turtle (*Emys orbicularis galloitalica*) Hatchlings in a Controlled Environment

**DOI:** 10.3390/ani10091510

**Published:** 2020-08-26

**Authors:** Simone Masin, Luciano Bani, Davide Vardanega, Norberto Chiodini, Valerio Orioli

**Affiliations:** 1Department of Earth and Environmental Sciences, University of Milano-Bicocca, 20126 Milano, Italy; simone.masin@unimib.it (S.M.); dade.varda93@gmail.com (D.V.); valerio.orioli@unimib.it (V.O.); 2Department of Materials Science, University of Milano-Bicocca, 20126 Milano, Italy; norberto.chiodini@unimib.it

**Keywords:** agonistic behavior, reptiles, Elo-rating, rearing, reintroduction project, hierarchy stability, straight carapace length, dyadic interactions, hierarchy steepness

## Abstract

**Simple Summary:**

Many animal societies are organized in hierarchical structures, which are determined by behavioral interactions among individuals. However, the study of complex social relationships in reptiles, particularly in turtles, is still poorly studied and usually performed on adult individuals. We investigated the establishment of hierarchical social structures in hatchlings of the European Pond Turtle during their first year of life. We reared three small groups of turtles in a controlled environment and observed their pairwise interactions during food supply, on a daily basis for seven months. Data analysis was aimed at identifying the eventual hierarchical structures and describing their characteristics. The hatchlings started interacting at two months old and all groups established stable hierarchies after about one month of interactions. Turtles interacted by head bites, tail bites and mounts, but the effectiveness of these behaviors in establishing the rank of individuals was low. Both the interactions’ outcome and social ranks were independent from the turtles’ size. This study is first proof of the emergence of a social structure in the hatchlings of the European Pond Turtle in captivity, a condition that is often faced in ex situ conservation projects for this protected species.

**Abstract:**

Few species of reptiles are known to establish stable social structures and among these, chelonians provided scarce and conflicting results. Moreover, studies on turtles are usually performed on adult individuals. In this study, we checked whether and when hatchlings of the European Pond Turtle (*Emys orbicularis galloitalica*) established stable hierarchical structures in their first year of life, whether hierarchies were stable in time and how steady they were. We also verified whether social ranks were associated to the individuals’ size. We observed dyadic interactions daily within three small groups of turtles reared in a controlled environment for seven months. After two months, the hatchlings started to interact and progressively established stable hierarchical structures. However, the effectiveness of the three types of observed aggressive behaviors in reversing social ranks was low and the resulting hierarchies were flat. We did not find a significant effect of the turtles’ size on their interactions’ outcome and hierarchy structure. Our results provide clear evidence of the development and the characteristics of social behaviors in young reptiles in captivity. This study could be a starting point for investigating social structures in wild populations.

## 1. Introduction

Until the second half of last century, the current view about reptile social behaviors envisaged this class of vertebrate as almost unfit to build and manage long lasting and complex social relations, like taxa with comparable or even lesser cognitive complexity do, such as eusocial hymenopteran, amphibians or fish [1,2]. The first works that called into question this reductionist approach were from Burghardt et al. [3]. They observed and described social behaviors in Green Iguana (*Iguana iguana*) hatchlings simultaneously hatched in the same reproductive site on a small island. Iguana hatchlings were able to synchronize nest emergence in order to migrate together, reducing the risk of predation. In addition, several social interactions (e.g., reciprocal licking, mounting behavior) more frequent among siblings, were found to be important for group cohesion. Following studies, mainly run on snakes and lizards, showed that an increasing number of species were able to produce social behaviors aimed to the establishment of within-group hierarchies, often used to regulate the access to resources. In Burmese Python (*Python molurus bivittatus*) and Diamondback Rattlesnake (*Crotalus atrox*), groups of males establish a linear hierarchy that regulate the access to the females during the reproductive season. In these species, a significant relationship between the individual rank and the number of successful copulations was found [4,5]. Studies on the lizard *Uta stansburiana*, a highly social species showing complex breeding strategies, with three male phenotypes, each one correlated with a different mating strategy, disproved the traditional view of reptiles as animals not suited for complex social interactions [6,7]. According to Gardner et al. [2], the definition of sociality in reptiles cannot be separated from issues of seasonal and spatial stability. A fundamental precondition for the development of stable social bounds and complex relationships is the sharing of spaces and activities by a stable group of individuals. In a review on this topic, Gardner et al. [2] found suitable criteria of “stable social aggregation” in less than 20% of the 94 reptile species classified as showing social aggregations cases. In such a general framework, chelonians represent a sort of grey area. Studies focused on social interactions and stable social structures are few and usually limited to dyadic interactions between pairs of individuals, more centered on the study and description of agonistic behaviors than on a comprehensive view of possible hierarchical structures in the group itself [8]. A number of works on colonies of turtles and tortoises in captive conditions seems to show that a few species are able to establish linear hierarchies, which are not always related to biometric parameters or to reproductive success [9]. Even more lacking seems to be the knowledge about the early establishment of hierarchies amongst juvenile or hatchling individuals, in spite of the evidences that these animals are able to form stable aggregations as adults, with fission–fusion group dynamics [10].

On these premises, we planned a study of the European Pond Turtle (*Emys orbicularis galloitalica*) hatchlings, with the purpose to study the possible presence and the early onset of hierarchies in hatchlings and juveniles of the species in a controlled environment. The studied individuals were captive reared for a reintroduction project, which entails a head-starting protocol (authorized by Lombardy Regional Administration, Directorate General for Environment and Climate, #T1.2016.0007767). We sought to answer the following research questions:Do hierarchies exist in same-age groups of European Pond Turtle hatchlings? Which are the most effective behaviors individuals adopt to establish and maintain hierarchies?If hierarchies are present in juvenile turtles, when does a group of individuals establishes such a hierarchy? Is this hierarchy stable over time? How steady is it?Is the rank of individuals related to one or more biometric parameters (e.g., size, weight)?

## 2. Materials and Methods

### 2.1. Rearing Set-Up

For this study, we acquired 16 European Pond Turtle hatchlings, 5–10 days old, from an authorized breeding facility (Oasi di Sant’Alessio, Sant’Alessio con Vialone, Pavia, Italy) for this endangered species (Annex II of Habitat Directive 92/43/EEC, near threatened in Europe [11], and endangered in Italy [12]). All the individuals were hatched in September 2017 from semi-natural incubation (fenced area without control of temperature and hygrometry) of the eggs, laid by a captive population hosted in the facility. No data were available on the sex and clutch of origin of individuals.

The hatchlings were reared within three breeding units, using plastic tanks (base 100 × 60 cm, height 20 cm). To avoid any potential overcrowding problem, starting from the beginning of the rearing protocol, the hatchlings were divided into small groups (5, 5 and 6 for the three tanks, respectively) by means of a randomized procedure and were never mixed during the study period. We connected each unit with an external filter device (maximum water flow rate: 650 L/h, filtering volume: 3 L). We used a dual-purpose (biological-mechanical) canister filter, equipped with ceramic units and a synthetic sponge. We maintained maximum water depth at around 10–15 cm for the entire breeding period, in order to avoid drowning casualties among the hatchlings, and we heated the water with an aquarium heater set at 27 °C. We equipped each breeding unit with an emerged basking area (dimension 30 × 10 cm) made of building bricks. Each basking area was equipped with a 5% emission UV-b lamp (Zoomed, 25 W) and a basking filament lamp (60 W), maintaining the basking area temperature at 30–35 °C. The photoperiod was fixed to 12–12 h. We made partial water changes on a weekly schedule. During the first month of life, we fed the hatchlings with *Chironomous* sp. larvae and turtle baby pellets (JBL Baby Formula, JBL GmbH & Co. KG, Neuhofen, Germany). Thereafter, we fed the animals with fresh food―thawed raw fish (minnows, *Atherina boyeri*), mealworms (*Tenebrio molitor*) and crayfish tails; dry food―turtle pellets (sera Raffy Mineral and sera Raffy P, sera GmbH, Heinsberg, Germany), minced vegetables and cuttlefish bone. We fed the turtles ad libitum once a day in the morning, for a period of ten minutes per day. Possible food remains were removed from the tanks.

### 2.2. Biometric and Behavioral Data

For the behavioral experiment, we marked the 16 hatchlings on the carapace, with a code for an easy identification at a 1 m distance. We monitored the hatchling growth daily by measuring two distinct biometric parameters: straight carapace length (SCL) and the weight of fasting individuals (W). For this purpose, we used an analogue vernier dial caliper (±0.1 mm) and an electronic scale (±0.1 g).

We systematically collected interaction data using a standardized protocol five days per week, from November 2017 to June 2018. We started to collect behavioral data from the end of the second month of age, when we observed the onset of the first social interactions. We fasted the animals for at least 20 h before each test. Then, we stimulated agonistic interactions by placing a piece of food at a time in the middle of the tank. We recorded the dyadic interactions (i.e., between two individuals) that spontaneously arose without an a priori forced selection of the interacting pairs in order to reflect individual propensity to interact. Two independent observers recorded (i) the individuals involved in the interaction; (ii) the dyadic interaction observed; (iii) the outcome of the interaction (i.e., which subject resulted dominant, which subject started/ended the interaction). Each observation session had a 5 min duration. We classified the dyadic interactions as: (a) head bite (HB), when the dominant individual bit the front parts (mouth, head or forelegs) of the dominated, which in turn hid the head and forelegs into the carapace; (b) tail bite (TB), when the dominant individual bit or attempt to bite the tail or hind legs of the dominated, which in turn hid the tail and legs into the carapace or fled; (c) mount (M), when the dominant individual mounted the carapace of the submissive individual, often grasping the carapace with both forelegs; the two individuals engaged in this interaction could remain bound for a few minutes, or the dominated individual could alternately try to swim away from the dominant. Seven students were involved as observers and were trained by the authors (SM and NC). Inter-observer reliability was assured by authors both during training and by random controls.

### 2.3. Data Analyses

A plethora of methods to infer dominance hierarchies from agonistic dyadic interactions currently exists [13,14]. However, most of them are matrix-based approaches (e.g., David’s score, I&SI and ADAGIO) that assume a stable dominance hierarchy and were proved inadequate to describe hierarchy dynamics [13,15]. Alternatively, a sequence-based method, such as the Elo-rating [16], and the inferred versions of matrix-based methods were efficiently used to track the changes of individuals’ rank over time [15]. Although different upgrades of these methods are available, we considered the original Elo-rating the most appropriate to detect the presence and to track the dynamic of an eventual dominance hierarchy in the turtle hatchlings’ data. We discharged the randomized Elo-rating, because it assumes the sequence of interactions to be uninformative, and the inferred versions of the Elo-rating and matrix-based methods, because they require information about prior knowledge of dominance correlates and the subdivision of the study period into biologically meaningful time intervals (parameter p in [15]). Both parameters are unknown for the European Pond Turtle or close related species, whose social structure has never been studied before, particularly during the hatchling period.

The Elo-rating is a rating-based method originally proposed for ranking chess players and subsequently applied to many animal behavioral studies (e.g., [17,18,19,20]). It is calculated as an individual score that updates sequentially after every dyadic interaction involving the individual themselves. The score of the winning and losing individual increases or decreases, respectively, according to the outcome of the interaction by an amount proportional to the difference between the expected and the realized outcome. The expected outcome depends on the difference in rating between the two participants before the interaction occurs, with higher expectations corresponding to a higher difference in rating. Thus, highly expected outcomes (a high-ranked individual defeating a low-ranked individual) lead to low changes in the Elo-score and vice versa [21]. The speed of rating change is controlled by the parameter k, which can be constant or can vary according to the intensity of the interactions, the contested resource value or the species-specific ability of participants to gain experience from previous challenges [16,21]. The choice of k is awkward when the mechanisms underlying the studied hierarchical society is unknown, but statistical methods to find a posteriori the best k value and the determinants of k variation have recently been proposed [22,23]. Similarly, the starting value assigned to each individual can disentangle individuals or a group of individuals having a different status at the beginning of the study, which could have arisen from an already established hierarchy. When initial individuals’ ranks are unknown, the starting values can be set equal for all individuals or can be estimated from a large range of possible values through an optimization procedure [23].

The sequential calculation of the Elo-rating can be affected by temporal bias due to variations in the individuals’ probability of interaction or in the behavioral observation process [13]. We avoided any bias by not introducing new individuals, nor preventing any individual’s interaction with the others within groups, and by adopting a constant standardized data collection procedure during the whole observation period.

We analyzed (VO and DV) the three groups separately and we calculated the individual Elo-ratings from the daily sequence of the observed dyadic interactions, to detect the presence of dynamic hierarchies (Research question 1). We then compared two models, one with a constant k and one with interaction-dependent k values, in order to test the strength of different agonistic behaviors in ensuring winning and losing probabilities. We used the optimization procedure implemented in the EloRating package v. 0.46.10 [23] in R environment v 3.4.0 [24], looking for the best k within the range 1–200 (resolution: 200) for the one- and the three-interaction (M, HB, TB) model. We set the initial score of every individual at 1000 [21]. We then compared the two models by calculating the differences between four maximum log-likelihood values: one for the one-interaction model and one for each interaction of the three-interaction model, when the values of the other interactions were set at their optimal value. We also compared the models by visually inspecting the four curves describing the variation of the log-likelihood in function of k. To check for the existence of an unobserved previous hierarchy, we built a model with a different starting value for each individual, by testing 2000 starting values randomly selected within the range of 200 standard errors around the value 1000 (*EloRating* package). We finally compared the best k-optimized model with the starting value-optimized model by calculating the difference in the models’ log-likelihood.

To better describe the hierarchies’ dynamics and to identify the timing of the hierarchies’ settlement (Research question 2), we evaluated the temporal variation of hierarchy stability and the steadiness of hierarchies, i.e., the degree of hierarchies’ steepness. We assumed that an abrupt increase in stability and steepness should be observed, if hierarchies were established after a cutoff in the hatchlings’ growth. We assessed the hierarchy stability through the computation of the modified stability index S [25], a measure of the frequency of rank changes that ranges from 0 (unstable) to 1 (highly stable), for each of 30 weekly intervals we split the study period into. Conversely, we assessed hierarchy steepness by splitting the study period in seven intervals, by adding one month at a time from the starting date. As a measure of hierarchy steepness, we derived the shape of the curve of the probability of winning for the higher ranked individual respect to the rank difference between the two individuals involved in a dyadic interaction [13] from the best Elo model of the three groups [23]. We then compared the variation of the curve shape between the increasing-time periods using the package *aniDom* v 0.1.3 [26].

Size is weakly correlated to age in reptiles [27], as body growth decelerates in most species after sexual maturity is reached [28] and many other factors interact to shape aging-related life-history traits [29,30]. In addition, the size–age scaling can be artificially mismatched when body growth is manipulated in experimental [31] or head-starting projects [32]. To test the effect of growth on the hatchlings’ social behavior (Research question 3), we thus investigated the role of straight carapace length (SCL) and body weight (W) in shaping hierarchy structure at two temporal scales. First, we tested whether higher ranked individuals were larger in size and weight at the (daily) interaction level, expecting that body size difference positively correlated with the Elo-score difference between the two contestants. Second, we checked whether size differences determined hierarchies in the long term, i.e., whether the hierarchies’ ranks matched the size ranks on a weekly basis. For both the analyses, we built a generalized linear mixed model using the *lmerTest* package [33] in an R environment. In the daily model, we regressed the difference in Elo-score before each interaction on the difference in SCL between the winner and the loser contestant, allowing winner identity to vary across intercept and slope as a random factor. In the weekly model, we regressed the difference in the Elo-score of each individual with respect to the higher ranked individual on the difference in the SCL of each individual, with respect to the longest individual, measured at the end of each of the 30 weekly intervals used in the previous analyses. Individual identity was included as a random effect varying on intercept and slope, as in the daily model. All the analyses were run for the three hatchlings’ groups and with SCL or W as a predictor, separately.

The R code for all the analyses is available upon request. Raw data are openly available from Figshare (dx.doi.org/10.6084/m9.figshare.12746510).

## 3. Results

We observed 712, 910 and 775 dyadic interactions in Group one, two and three, respectively. No unknown interactions were observed in any of the groups, nor draws. The number of dyadic interactions per individual ranged from 6 to 78 in Group one, from 7 to 91 in Group two and from 7 to 113 in Group three (Figure 1). HBs accounted for 67–80% of the dyadic interactions, while Ms ranged from 15% to 29% and TBs were always lower than 4%. The three behaviors were observed in a similar proportion during the whole study period.

Disentangling interaction intensity and a priori entry ranking did not give a better prediction of the winning probabilities than the model with constant k and starting value (set to 1000). Indeed, the model with interaction-dependent k values did not outperform the constant-k model for any group. The maximum Log-likelihood of the interaction-dependent model of Group one was −473.6 (best ks = HB: 12, M: 18, TB: 5), while it was −476.5 for the constant-k model (k = 11, Appendix A). The maximum Log-likelihoods for Group two (Appendix A) were −625.0 (best ks = HB: 26, M: 9, TB: 6) and −625.8 (k = 6) for the two models, respectively, and they amounted to −493.6 (best ks = HB: 5, M: 18, TB: 9) and −494.8 (k = 12), respectively, for Group three (Appendix A). Optimizing starting values resulted in equally performing models, whose maximum Log-likelihood was −475.7, −625.0 and −492.4 for the three groups, respectively. We thus showed the results of the most parsimonious model, with constant k and starting value, and we used it for the following analyses. Detailed results of the optimized models are reported in Appendix A.

Elo-scores’ dynamics highlighted that two individuals (ID 13 and 14) alternatively dominated the hierarchy of Group one, while the other four individuals were often defeated (Figure 2a). In Group two, one individual dominated the others for most of the study period (ID 4), but in the final months individuals 2 and 1 recovered some rank in the hierarchy. Individual 9 was constantly the lowest-ranked individual (Figure 2b). Group three showed the clearest dynamic, as no rank change occurred from March onward. Here, individual 5 dominated the group, while individuals 6 and 8 occupied the lower ranks (Figure 2c).

The hierarchy of the three groups of hatchlings reached a high stability after a first period of adjustment, as the weekly stability index settled above 0.85 two weeks after the start of observations. Particularly, the stability index ranged between 0.91 and 1.00 from December onward in Group one and it was chiefly higher than 0.95 in the other groups, except for two negative peaks at mid-April (0.92) and mid-May (0.87) in Group two and a negative peak in February (0.86) in Group three (Figure 3). These results reflect the relatively low rate of rank change in all three groups during the study period (Appendix A).

The hierarchy steepness differed between the three groups, although it did not substantially increase in time in any group. In Groups one and two, the probability of the dominant individual winning slowly increased with the difference in rank to the subordinate and did not clearly change in time (Figure 4a,b), except for the shape of the relationship of June in Group one that was steeper. The effect of the rank difference was stronger for Group three, where the probability of the higher ranked individual winning an interaction differed by about two decimals between one and four rank difference. In this group, a slight increase in winning probability in time was observed (Figure 4c).

The effect of growth of the hatchling turtles on their social behavior resulted in contrasting patterns. W was highly correlated to SCL (Pearson r = 0.978; *p*-value < 0.001; N = 2449) and only the SCL results are presented hereafter. SCL quickly and constantly increased during the study period, consistently among individuals (Appendix A). The average SCL grew from 3.75 cm (SD = 0.49) to 6.75 cm (SD = 0.67).

In Group one, the larger the differences in SCL, the more negative the difference was in the Elo-score between the two contestants (Figure 5a, Table 1). The effect was consistent among individuals as the standard deviation of the random effect for the slope was relatively low (Table 1). This pattern was strongly influenced by individual 7, which was bigger than the others (Appendix A) but occupied the lowest ranks of the hierarchy (Figure 2a) and ID 13, which was the smallest individual but was the second ranked in the hierarchy. In Group three, we observed a similar result (Figure 5c), but a stronger negative relationship between the SCL difference and Elo-score difference with a higher variability among individuals (Table 1). Conversely, in Group two, the difference in SCL increased with larger difference in the Elo-score (Figure 5b). The *p*-value of the average regression coefficient of SCL difference was slightly above the threshold of 0.05, as a high variability (standard deviation) of slope among individuals was observed (Table 1).

The difference in SCL from the larger individual did not correlate with the difference in Elo-score from the higher ranked individual on a weekly time frame for any of the three groups (Appendix A). At this time scale, the variability of the relationship was very high and inconsistent among individuals (see random effects in Appendix A).

## 4. Discussion

The establishment of hierarchies in chelonians is still a debated topic, since a number of studies seem to point out that durable hierarchic structures are present among adults in some species of turtles [9,34] and that in some cases, these hierarchies show a linear feature. However, the onset of these social structures in groups of hatchlings or juvenile individuals is somehow less studied, so the question about the dynamics of development of hierarchic organizations in very young individuals remains unanswered [35]. This lack of knowledge could be due to both the scarcity of data about the first stages of the life cycle (from the stage of hatchling to the stage of juveniles [28]) and to the complexity of the experimental set-up to evidence the early onset of hierarchical structure in very young turtles and tortoises [34,35].

According to our results, in European Pond Turtle juveniles early hierarchic structures seem to appear among individuals of the same group after around two months of age. Before that time, the agonistic behaviors between two individuals were scarce and inconspicuous. Although studies about the early onset of hierarchic structures and agonistic–aggressive behaviors in very young reptiles and precocial birds seem to portrait a diversified scenario [35,36,37], general evidences confirm the idea that the onset of hierarchies and the formation of stable hierarchic structures is not immediate after the hatching. For example, domestic chickens (*Gallus gallus domesticus*) may begin to interact after 4–5 days after the hatching, but the onset of aggressive behaviors can be delayed until the birds are 15 days old [38]. Studies in juvenile Snapping Turtles (*Chelydra serpentina*) pointed out that hierarchies can be observed after four months [35], suggesting that the formation of stable social interactions for food competition may be of rather widespread occurrence even among quite unrelated chelonians. Once established, however, the hierarchies among juveniles of the European Pond Turtle in our study seem to be relatively stable in time, with a low steepness. Indeed, the result of a single aggressive interaction seem not so decisive in the process of changing the rank of the two individuals involved in the dyadic interaction. Even the non-linearity of the social structure established among individuals could reflect the husbandry parameters. Although linear hierarchies among social species are witnessed in wild populations, these structures seem to be more frequent in captive situations, since many species in the wild form fission–fusion societies, rather than stable groups, thus allowing non-linear hierarchic structures [39].

The comparison among the different agonistic interactions and their efficiency in determining a change in the rank of the winner evidenced that all interactions seem to have the same efficiency in determining a change in the rank status of the individual that won a dyadic interaction. This evidence is contrasting with the work of Bush et al. [40] on adult males and females of Green Anoles Lizards (*Anolis carolinensis*), which pointed out that some type of behavior is more effective than others in determining the success of an individual. This could be due to the different complexity of social structures in Anoles Lizards compared with turtles, and to the young age of the individuals we studied. Concerning this particular point, it is also important to stress that the relative rank of individuals seems to be unrelated to their size. Indeed, some small-size juveniles were nonetheless able to win multiple interactions with bigger individuals and conquer and maintain a higher rank position. It is well known, however, that in some animal species the size/weight is not an accurate predictor of the actual social rank of a given individual [40].

We must point out some considerations about the environmental parameters and husbandry practices that could have introduced elements of disturbance in this study. The first consideration is about the food supply schedule. The ad libitum feeding practice (although made for clear conservation purposes, in order to achieve the maximum growth speed of the stock) could have led to a low intensity of agonistic interactions in all the experimental groups, thus blurring the hierarchic order and making the onset and maintenance of social structures less evident, as well as the steepness of the hierarchy itself. In a similar way, the density of individuals in each group in relation to tank dimension could influence the interactions among individuals. We based the evaluation of the minimum space required per individual on previous studies about this species in captive conditions [41], but the effects of captive rearing conditions on social interactions of *Emys orbicularis* hatchlings are seldom studied. Again, concerning the latter point, it should be considered that while the social structures of adults in European Pond Turtle are well known from wild populations, those of hatchlings and juveniles are not clear ([42]). Since the mean number of eggs per nest is small and the hatchlings are very tiny and inconspicuous until they are 4–5 years old, surveys of hatchlings and juveniles’ aggregations in the wild are difficult to perform and, consequently, young turtles’ social behavior in the wild is scarcely known [9]. Further studies in young individuals of European Pond Turtle in the wild will shed light about the conditions (e.g., population density, food availability) that could affect the emergence and intensity of social structures. Finally, the time interval considered could as well have had influenced the outcome of this study. A longer observation period could have produced a better understanding of the stability of hierarchies, or shed light upon the social changes in the hierarchy as the juveniles grow.

We also acknowledge some strengths and limitations of the analytical framework that should be considered when interpreting the results or comparing them to other studies. The methods’ performance was proved to depend on the sparseness of the dataset (i.e., the unequal distribution of interactions per individual) and on hierarchy steepness (i.e., the shape of the relationships between the probability of the higher rank winning and the difference in rank between the interacting individuals). On one hand, our data were not sparse, as we collected a high number of dyadic interactions, and we yielded a very high ratio between the number of interactions and individuals (119, 182 and 155 for the three groups, respectively). These values corresponded to a low proportion of unobserved dyads and a low level of sparseness, according to a Poisson process [13]. On the other hand, the inferred hierarchies were very flat and this could have affected the estimation of individuals’ ranks and the following analyses that depend on them, i.e., the Elo-rating method, was proved to perform badly with low steepness. However, the benefit of the high number of observed dyads should have balanced the uncertainty due to a flat hierarchy [13]. Moreover, the standardized rearing protocol should have reduced the effects known to commonly increase the uncertainty in inferring Elo-scores, such as the balance and the size of the sample, the stability of the hierarchy, and the occurrence of changes in the hierarchy due to the immigration and removal of individuals, or to the presence of coalitions of individuals [43].

## 5. Conclusions

In conclusion, the outcomes of this study can contribute to the knowledge of the early onset of hierarchies among very young turtles and, in a more general sense, provide a better understanding of the complexity of reptile social structures, a topic often underestimated and neglected. The suggestions from this preliminary work could be useful for further studies on agonistic interactions of wild populations of young individuals of European Pond Turtle, as well as for the evaluation of the effectiveness of communal rearing conditions in captivity, in order to achieve the best possible condition for an optimal growth for ex situ conservation projects [44,45]. Further developments of this study can include variations in the experimental design as previously highlighted, i.e., variations in the quantity of given food per day, or the timing of food supply, in order to achieve a stronger motivation in the interacting subjects. Another point of further investigation could be the alteration of other environmental parameters (typically, water temperatures), in order to correlate physiological manipulations with the intensity of the agonistic behaviors and the stability of the hierarchies. In addition, the importance of other resources in establishing hierarchies could be investigated, especially when limiting factors act simultaneously. For example, especially for pond turtles, the availability of basking sites, together with food supplies, could be of particular relevance [46,47,48], and their combined effect may strengthen or alter the final hierarchical structures.

## Figures and Tables

**Figure 1 animals-10-01510-f001:**
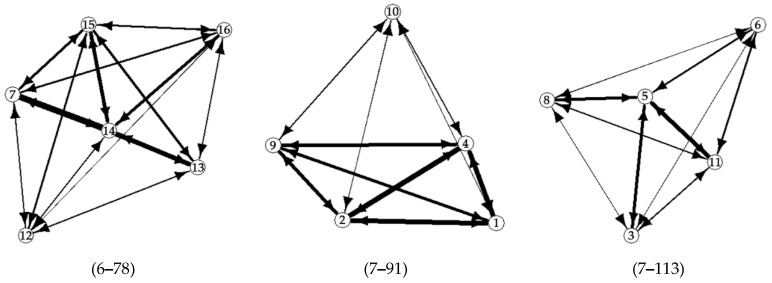
Dyadic interactions observed in the three groups of European Pond Turtle hatchlings along the seven months study period. Nodes’ numbers indicate the identity of the hatchling turtles, arrows indicate the direction of interactions and the line thickness is proportional to the total number of observed interactions within the group (the range of interactions are in brackets); the hidden arrow between individual 7 and 13 equals to 49 interactions.

**Figure 2 animals-10-01510-f002:**
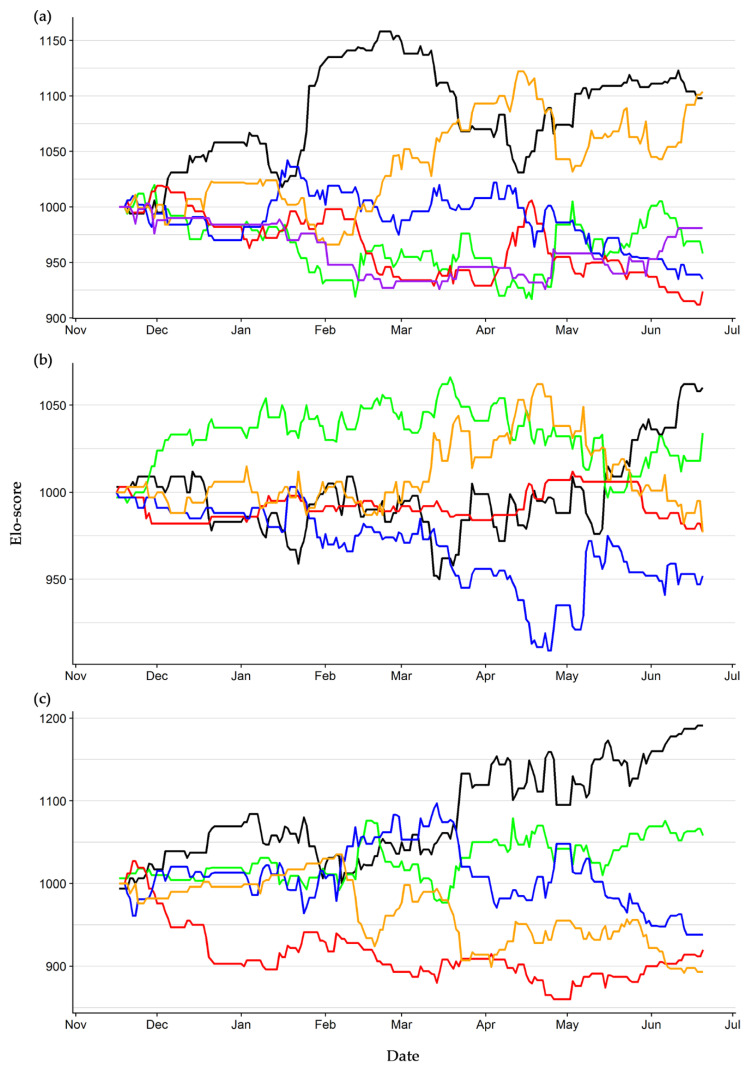
Elo-score dynamic of the hatchling European pond turtles reared in three separate units from November 2017 to June 2018. Group one (**a**): ID 7 (green), 12 (purple), 13 (black), 14 (orange), 15 (blue), 16 (red). Group two (**b**): ID 1 (orange), 2 (black), 4 (green), 9 (blue), 10 (red). Group three (**c**): ID 3 (green), 5 (black), 6 (red), 8 (orange), 11 (blue).

**Figure 3 animals-10-01510-f003:**
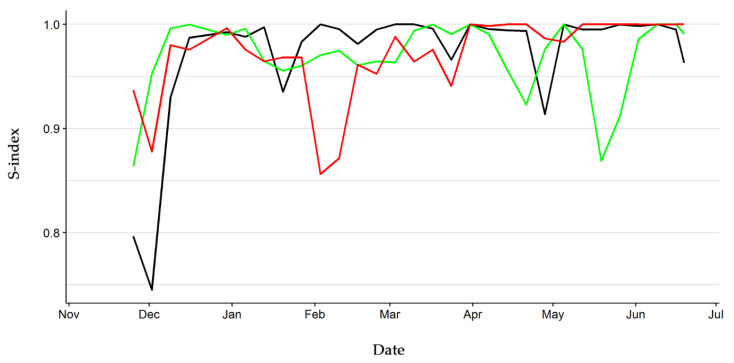
Hierarchy stability index of the three groups of hatchling European pond turtles. Black line: Group one. Green line: Group two. Red line: Group three.

**Figure 4 animals-10-01510-f004:**
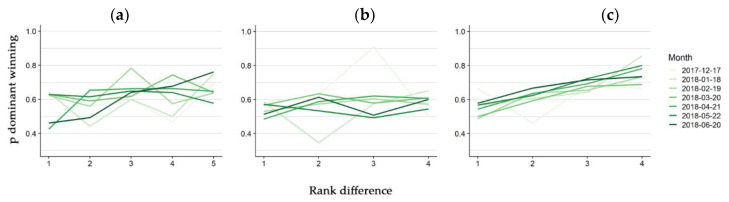
Hierarchy steepness variation in time. Monthly probability of the dominant winning (*y* axis) a dyadic interaction in the function of the difference in rank to the submitted individual participating in the interaction for Group one (**a**), two (**b**) and three (**c**). One line for each month was fitted and it was plotted with increasing green intensity.

**Figure 5 animals-10-01510-f005:**
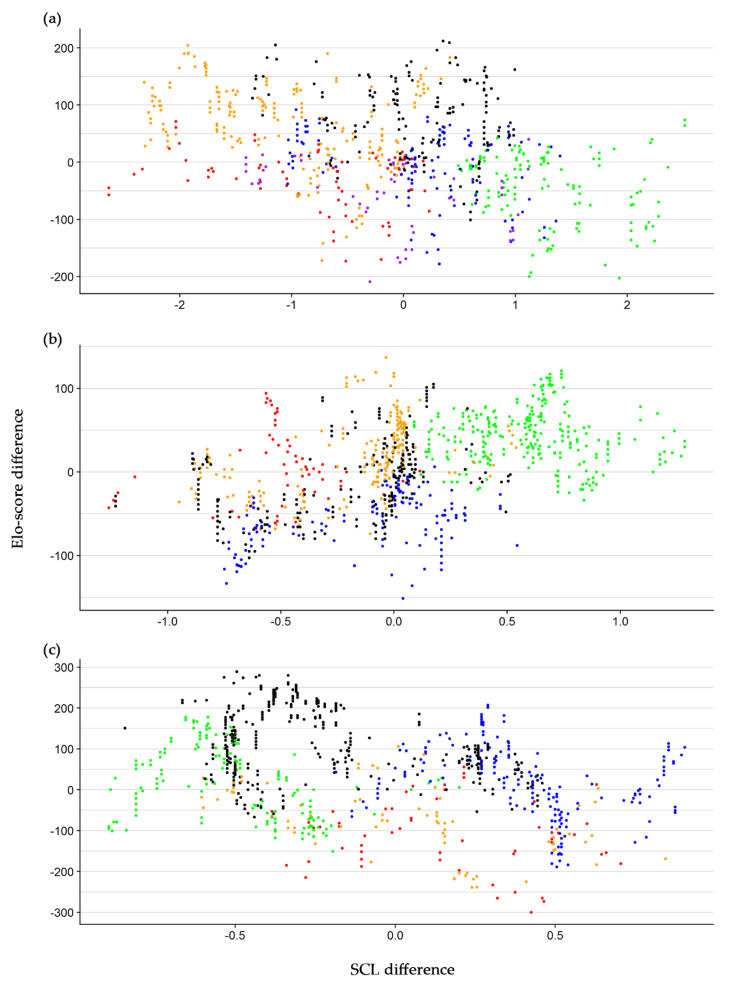
Relationship between the difference in Elo-score and the difference in straight carapace length (SCL) before the interactions occurred. Each dot represents a single dyadic interaction and its colors identify the winning individual. Group one (**a**): ID 7 (green), 12 (purple), 13 (black), 14 (orange), 15 (blue), 16 (red). Group two (**b**): ID 1 (orange), 2 (black), 4 (green), 9 (blue), 10 (red). Group three (**c**): ID 3 (green), 5 (black), 6 (red), 8 (orange), 11 (blue).

**Table 1 animals-10-01510-t001:** Linear mixed models of Elo-score difference on the SCL difference at the interaction level for the three groups of hatchling European Pond Turtles. The identity of the winning individual was used as a random effect for both the intercept and SCL Difference; SD: standard deviation of the random effect.

Group	Intercept			SCL Difference			SD of Random Effect	
	Value	SE	*p*-Value	β	SE	*p*-Value	Intercept	SCL Difference
1	−1.70	21.77	0.941	−26.92	3.49	<0.001	52.95	1.48
2	7.61	16.41	0.667	31.72	12.89	0.068	36.34	26.95
3	−10.36	36.96	0.793	−97.94	10.22	<0.001	82.27	5.12

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
