# Peer review of "Hierarchies and Dominance Behaviors in European Pond Turtle (Emys orbicularis galloitalica) Hatchlings in a Controlled Environment"

_animals, 2020, doi:10.3390/ani10091510_

Round 1

Reviewer 1 Report

I have read the manuscript by Simone Masin and colleagues, submitted to the journal Animals. The manuscript describes study on hatchlings of the European pond turtle Emys orbicularis galloitalica in captivity, the study on hierarchical structures. Data on social structures among chelonians are scarce. I think, the data presented in the manuscript are worth to be published. However, I think the version of the manuscript could be improved. I hope that the following comments can be used to improve the manuscript.

General comments:

One of the important factor (analysed in the manuscript) is effect of size of the turtles on hierarchical structures. In the manuscript, there are information that hatchling growth (i.e., size and weight) was measured daily, but no data about size are in the text. It could be important, for example, differences in size between the smallest and largest hatchlings in each of groups (for example, on the beginning and the end of the study); I see that it is possible to find such data from supplementary materials, but for readers some information in the text would be useful.

I do not understand why if (lines 63-65) “fundamental precondition for the development of stable social bounds and complex relationships is the sharing of spaces and activities by a stable group of individuals during the year.”

but Authors studied the hatchlings for seven months (e.g., lines 18-19).

I am not sure, if I understand the protocol. “We systematically collected interaction data using a standardized protocol five days per week (…)” (line 118), but (lines 121-122) “Then, we fed them with a single piece of food at a time, placed between two individuals.”

So, if each(?) pair of animals was studies, e.g., every week? If the pairs were randomized etc.? [see also below, comments to Fig. 1: What about interactions between individuals number 7 and 13?]

There are some good papers on interactions among turtles, especially basking, adult individuals. Some of them are about the European pond turtle (however, mostly about competition between the European pond turtle and the introduced red-eared slider). See, e.g., papers by Cadi A. and Joly P. – the papers are not cited in the manuscript. I understand that it is not possible (and not necessary) to cite all papers on any subject, however, I think that citations, mostly in Discussion, should be checked. For example, lines 369-371: “...social structures of adult in European Pond Turtle are well known from wild populations, those of hatchlings and juveniles are not clear ([42])” – I have found no information about ‘social structures of adult’ in European Pond Turtle in the publication No. 42.

Specific comments:

lines 128 and next (the Mount behavior). Are you sure, that the “Mount” behavior is similar interaction to “Head bite” and “Tail bite”? (in the text the Mount behavior is entered ‘between’ the two typical aggressive behaviors) lines 126 and next:

-- (a) Head bite, …

-- (b) Mount, …

-- (c) Tail bite, …

Additionally: if the Mount behavior was observed in similar proportion during all the seven months? I know, that for hatchlings of Emys, such behavior (i.e., the Mount) could be present during shorter time (and sometimes looks like copulation behavior, no typical aggressive behavior).

I suppose that data on sex of the studied hatchlings, as well relationship (if they were from different clutches?), are not available. However, as it could be important, I think that short information about lack of such data should be included in the manuscript; in the Materials and method section, or in the Discussion section.

lines 201-202: “Size is weakly correlated to age in reptiles [27] as body growth decelerate in most species after sexual maturity is reached [28]” – yes, it is true, but you studied really young turtles (just hatchlings); I think that for such animals size is strongly correlated to age. So, I am not sure if the information is necessary here (i.e., in the Materials and methods section).

Fig. 1

-- You used 16 individuals, divided into three groups (5, 5 and 6 individuals).

In figure 1 “Nodes’ numbers indicate identity of the hatchling turtles”. Why in the first group there are numbers ‘7’, ‘12’, ‘13’, ‘14’, ‘15’, ‘16’? It could be confusing for readers, I think [the easiest way is: the 1st group: numbers 1-6, 2nd: 7-11, 3rd: 12-16, I think].

-- It is rather difficult to compare the data with, e.g., data presented in Fig. 2.

-- “line thickness is proportional to the total number of observed interactions” – it will be good to add approximate number of such interaction (e.g.: ‘the thin line = ~… interactions’).

-- What about interactions between individuals number 7 and 13?

line 284 “W was highly correlated to SCL (Pearson r = 0.978)” – p value  and sample size should be showed also, I think. And the word “Weight” (not “W”) would better in the sentence.

Figure 5.

“Colors identify the winning individuals.” – Forgive me for my possible misunderstandings, but  I am not sure if the information will be correctly understand by readers.

General comment to table and figure legends:

In scientific papers, captions of figures and tables should be ‘self-explaining’, i.e., should provide sufficient information to the readers without looking for information in the text. Thus, I think, that the description should be improved. For example:

-- I think that information what is showed on the y-axis is necessary (see: figures S04-S12).

-- Fig. 4. A short information in the figure legend, what is “p dominant winning”, would be useful for readers.

Author Response

One of the important factor (analysed in the manuscript) is effect of size of the turtles on hierarchical structures. In the manuscript, there are information that hatchling growth (i.e., size and weight) was measured daily, but no data about size are in the text. It could be important, for example, differences in size between the smallest and largest hatchlings in each of groups (for example, on the beginning and the end of the study); I see that it is possible to find such data from supplementary materials, but for readers some information in the text would be useful.

REP: As suggested by the reviewer, we added the SD of each group, other the mean values of SCL

I do not understand why if (lines 63-65) “fundamental precondition for the development of stable social bounds and complex relationships is the sharing of spaces and activities by a stable group of individuals during the year.” but Authors studied the hatchlings for seven months (e.g., lines 18-19)

REP: WE CHANGED THE SENTENCE, REMOVING “DURING THE YEAR” SINCE THIS COULD GENERATE A MISUNDERSTANDING IN THE CONCEPT WE AIMED TO EXPLAIN.

I am not sure, if I understand the protocol. “We systematically collected interaction data using a standardized protocol five days per week (…)” (line 118), but (lines 121-122) “Then, we fed them with a single piece of food at a time, placed between two individuals.” So, if each(?) pair of animals was studies, e.g., every week? If the pairs were randomized etc.? [see also below, comments to Fig. 1: What about interactions between individuals number 7 and 13?]

REP: WE THANK THE REVIEWER, FOR HIS/HER COMMENT. INDEED, THE SENTENCE WAS NOT ENOUGH CLEAR, AND WE CHANGED IT IN ORDER TO BETTER EXPLAIN OUR PROTOCOL. WE ACTUALLY DID NOT STUDY EACH POSSIBLE DYAD, BUT ALL THE POSSIBLE DYADIC INTERACTION THAT AROSE WITHIN THE TANK (WITHOUT RANDOMIZING THE PAIRS OUTSIDE THE TANK).

There are some good papers on interactions among turtles, especially basking, adult individuals. Some of them are about the European pond turtle (however, mostly about competition between the European pond turtle and the introduced redeared slider). See, e.g., papers by Cadi A. and Joly P. – the papers are not cited in the manuscript. I understand that it is not possible (and not necessary) to cite all papers on any subject, however, I think that citations, mostly in Discussion, should be checked. For example, lines 369-371: “...social structures of adult in European Pond Turtle are well known / p from wild populations, those of hatchlings and juveniles are not clear ([42])” – I have found no information about ‘social structures of adult’ in European Pond Turtle in the publication No. 42.

REP: REFERENCE ADDED IN THE CONCLUSION SECTION.

FOR WHAT IT CONCERNS THE CITATION 42, MITRUS STUDIED YOUNG INDIVIDUALS AT AGE ONE, AFTER THEIR RELEASE IN NATURE. THUS, WE THINK THAT OUR CITATION IS PERTINENT RESPECT TO THE SENTENCE.

lines 128 and next (the Mount behavior). Are you sure, that the “Mount” behavior is similar interaction to “Head bite” and “Tail bite”? (in the text the Mount behavior is entered ‘between’ the two typical aggressive behaviors) lines 126 and next: -- (a) Head bite, … -- (b) Mount, … -- (c) Tail bite, …

Additionally: if the Mount behavior was observed in similar proportion during all the seven months? I know, that for hatchlings of Emys, such behavior (i.e., the Mount) could be present during shorter time (and sometimes looks like copulation behavior, no typical aggressive behavior)

REP: WE THANK THE REVIEWER. WE CHANGED THE ORDER ACCORDING TO HIS/HER COMMENT.

AS SUGGESTED BY THE REVIEWER, WE ALSO CHECKED FOR THE PERIOD LENGTH or trend FOR THE three BEHAVIOURS. WE DID NOT DETECT NEITHER PERIOD NOR TREND FOR ALL THE INVESTIGATED BEHAVIOR. WE ADDED A SENTENCE IN THE RESULT SECTION.

I suppose that data on sex of the studied hatchlings, as well relationship (if they were from different clutches?), are not available. However, as it could be important, I think that short information about lack of such data should be included in the manuscript; in the Materials and method section, or in the Discussion section

REP: WE THANK THE REVIEWER. WE ADD THE REQUESTED INFORMATION SPECIFIED ON HIS/HER COMMENT IN MATERIALS SECTION.

lines 201-202: “Size is weakly correlated to age in reptiles [27] as body growth decelerate in most species after sexual maturity is reached [28]” – yes, it is true, but you studied really young turtles (just hatchlings); I think that for such animals size is strongly correlated to age. So, I am not sure if the information is necessary here (i.e., in the Materials and methods section).

REP: ACTUALLY, THE LINK BETWEEN THE TWO SENTENCES WAS NOT ADEQUATELY JUSTIFIED IN THE FORMER VERSION. THUS, ACCORDING TO THE REVIEWER COMMENT, WE MODIFIED THE TEXT, IN ORDER TO BE CLEARER, IN THE MEANING WE AIMED TO GIVE TO THE SUBJECT MATTER.

Fig. 1 -- You used 16 individuals, divided into three groups (5, 5 and 6 individuals). In figure 1 “Nodes’ numbers indicate identity of the hatchling turtles”. Why in the first group there are numbers ‘7’, ‘12’, ‘13’, ‘14’, ‘15’, ‘16’? It could be confusing for readers, I think [the easiest way is: the 1st group: numbers 1-6, 2nd: 7-11, 3rd: 12, 16, I think]. -- It is rather difficult to compare the data with, e.g., data presented in Fig. 2. -- “line thickness is proportional to the total number of observed interactions” – it will be good to add approximate number of such interaction (e.g.: ‘the thin line = ~… interactions’). -- What about interactions between individuals number 7 and 13?

REP: THE ID NUMBER OF EACH INDIVIDUAL WAS ASSIGNED BEFORE THE COMPOSITION OF THE GROUPS. THESE NUMBERS WERE THEN USED FOR THE RANDOM EXTRACTION OF THE GROUPS. THEREFORE, THERE IS NO REASON WHY THEY ARE USED SEQUENTIALLY WITHIN GROUPS. THIS HIGHLIGHT THAT GROUPS WERE NOT IDENTIFIED SUBJECTIVELY.

WE THANK THE REVIEWER FOR HIS/HER PRECIOUS COMMENT ABOUT THE HIDDEN ARROW OF THE DYAD 7-13. WE MODIFIED THE FIGURE AND ITS CAPTION IN ORDER TO ACCOMPLISH THE ALL REVIEWER OBSERVATIONS CONCERNING THE FIGURE 1.

line 284 “W was highly correlated to SCL (Pearson r = 0.978)” – p value and sample size should be showed also, I think. And the word “Weight” (not “W”) would better in the sentence.

REP: THANK YOU FOR THE COMMENT. WE ADD THE REQUESTED INFORMATION. ACCORDING TO THE EDITORIAL RULES, WE USE “W” (AS PREVIOUSLY DEFINED) INSTEAD OF WEIGHT.

Figure 5. “Colors identify the winning individuals.” – Forgive me for my possible misunderstandings, but I am not sure if the information will be correctly understand by readers.

REP: IN ORDER TO BE CLEARER, WE CHANGED THE TEXT OF THE FIG. 5 CAPTION. THE NEW TEXT SHOULD MAKE MORE SELF-EXPLAINING THE CONTENTS OF THE FIGURE.

General comment to table and figure legends: In scientific papers, captions of figures and tables should be ‘self-explaining’, i.e., should provide sufficient information to the readers without looking for information in the text. Thus, I think, that the description should be improved. For example: -- I think that information what is showed on the y-axis is necessary (see: figures S04-S12). -- Fig. 4. A short information in the figure legend, what is “p dominant winning”, would be useful for readers.

REP: WE REVISED THE FIGURES’ CAPTIONS IN ORDER TO MAKE EACH FIGURE MORE SELF-EXPLAINING. FOR FIGURE 4, WE EXPLICATED THE LINK BETWEEN THE Y-AXIS AND ITS MEANING. WE ALSO MODIFIED THE CAPTIONS IN THE SUPPLEMENTARY MATERIALS.

Reviewer 2 Report

The study is very interesting, well conducted and correctly analyzed.

The authors must give the number of the administrative authorization to have performed experiment in protected species.

More information must be given on the conditions of eggs incubation:

What temperature ? Hygrometry ? All eggs incubated at the same time ? What was the age difference of hatchlings at the beginning of the experiment. How many mothers ? Why the mother effect was not studied as a random effect ?

The incubation conditions of eggs are of great importance for turtles. These points must be at least discussed.

Line 107: hutchlings
Correct to hatchlings

Line 111: We fed the turtles ad libitum once a day in the morning, for a period of ten minutes per day.

Food remain was removed from aquariums?

Author Response

The authors must give the number of the administrative authorization to have performed experiment in protected species.

REP: AT THE END OF INTRODUCTION, JUST BEFORE THE RESEARCH AIMS, WE GAVE THE NUMBER OF THE ADMINISTRATIVE AUTHORIZATION (ALREADY PRESENT IN THE FIRST VERSION OF THE MANUSCRIPT).

More information must be given on the conditions of eggs incubation: What temperature ? Hygrometry ? All eggs incubated at the same time ? What was the age difference of hatchlings at the beginning of the experiment. How many mothers ? Why the mother effect was not studied as a random effect ? The incubation conditions of eggs are of great importance for turtles. These points must be at least discussed.

REP: THE HATCHING TOOK PLACE IN NATURAL CONDITIONS, WITHIN A FENCED AREA. THIS MEAN, THAT TEMPERATURE AND HYGROMETRY HAVE NOT BEEN ARTIFICIALLY CONTROLLED. THE HATCHLINGS WERE RECOVERED JUST AFTER THEIR HATCHING. THE TEXT WAS CHANGED IN ORDER TO BETTER SPECIFY THE INCUBATION CONDITIONS

Line 107: hutchlings Correct to hatchlings

REP: CHANGED ACCORDINGLY

Line 111: We fed the turtles ad libitum once a day in the morning, for a period of ten minutes per day. Food remain was removed from aquariums?

REP: AS NOTED BY THE REVIEWER, THE FOOD REMAINS WERE REMOVED FROM THE TANKS. WE ADDED THIS INFORMATION IN THE TEXT.

Reviewer 3 Report

Review: Hierarchies and Dominance Behaviors in European 2 Pond Turtle (Emys orbicularis galloitalica) Hatchlings in a Controlled Environment

This manuscript describes the process of measuring social hierarchies in captive bred European Pond Turtles during their first year of life and in relation to size. The authors found that hierarchies existed with varying stability across groups and occurred independently of carapace length. The objectives are described clearly and are pertinent to the described methods and results. Although, I am not adequately versed in the application of Elo-scores to denote appropriateness of methods, the methods are clearly written with enough detail to ensure repeatability. The results are also clear, concise and easy to follow.

I would note that the outcomes of this study are explicitly limited to food competition and there are other important potential consequences for hierarchies in pond turtles. For instance, access to basking sites, which can also be a density dependent limiting resource was not quantified. It would be interesting to note if dominance in one realm of resource acquisition translated to other resources. For example, does a turtle that gets the food more also have greater access to the best basking location. Although the authors most likely do not have the data to include, it may be worth pointing out in the discussion that food is only one of several resources. Future studies could explore how much cross over there is in hierarchies across situations.  

Note minor grammatical issues: be most alert for use of has, had, have, as well as is and are throughout. Otherwise, the introduction and discussion could use some additional editorial work. I identified a number of issues below to assist the authors, although my comments are not comprehensive.

Simple Summary

LN15: I’m not sure that I would agree that the study of social relationships in reptiles is jeopardized. It is poorly studied.

LN18: Consider moving daily to after interactions.

LN20: Insert at before two months

L22: Effectiveness at achieving what? This needs to be defined.

Abstract

LN30: Does “how much steep they are” make sense? I would suggest the authors revise to improve clarity.

LN31: replace is with are.

LN 31 – 32: Suggest moving daily to after interactions

LN38: Based on the methods described, I do not belief much can be inferred about the turtles’ intent. Perhaps the authors were describing the application of the data towards informing future studies of turtle sociality in the wild? I would suggest revising if so.

Introduction

LN44-47: I would recommend integrating this into the first paragraph, as a stand along sentence usually isn’t an optimal way to start a paper. Also, the sentence is a bit of a run on and could benefit from being broken into two sentences.

LN61: Suggest replacing “definitely put an end to” with “disproved”.

LN64. Suggest starting a new paragraph with “According to Gardner…”

LN72: Should there be an ‘a’ in front of few? Is this sentence saying that species are able to establish them or that they are not? The second half suggests that they are, but the first half is negative in structure.

LN74: I do not understand what “state-of-the-art” means in the context of this sentence.

Methods

LN91: Replace “a set of” with sample size.

LN183: Replace visual with visually

Discussion

LN336: Replace ‘the’ with ‘they’?

LN360: Do the authors mean elements of disturbance?

LN362: Replace ‘has’ with ‘have’

LN368: Italicize species name.

LN384-387: This sentence is very hard to understand. I would suggest revising.

Author Response

I would note that the outcomes of this study are explicitly limited to food competition and there are other important potential consequences for hierarchies in pond turtles. For instance, access to basking sites, which can also be a density dependent limiting resource was not quantified. It would be interesting to note if dominance in one realm of resource acquisition translated to other resources. For example, does a turtle that gets the food more also have greater access to the best basking location. Although the authors most likely do not have the data to include, it may be worth pointing out in the discussion that food is only one of several resources. Future studies could explore how much cross over there is in hierarchies across situations.

REP: WE THAN THE REVIEWER FOR HIS/HER INTERESTING COMMENT. ACCORDINGLY, WE ADDED A COUPLE OF SENTENCES IN THE CONCLUSION SECTION.

Note minor grammatical issues: be most alert for use of has, had, have, as well as is and are throughout. Otherwise, the introduction and discussion could use some additional editorial work. I identified a number of issues below to assist the authors, although my comments are not comprehensive.

REP: FOLLOWING THE REVIEWER SUGGESTION, WE READ AGAIN THE TEXT IN ORDER TO CORRECT ALL TYPOS.

LN15: I’m not sure that I would agree that the study of social relationships in reptiles is jeopardized. It is poorly studied.

REP: CHANGED ACCORDINGLY

LN18: Consider moving daily to after interactions.

REP: CHANGED ACCORDINGLY

LN20: Insert at before two months

REP: CHANGED ACCORDINGLY

L22: Effectiveness at achieving what? This needs to be defined.

REP: ACCORDINGLY TO THE REVIEWER COMMENT, THE SENTENCE WAS MODIFIED IN ORDER TO BE CLEARER.

LN30: Does “how much steep they are” make sense? I would suggest the authors revise to improve clarity.

REP: WE REMOVED THE TERM “STEEP” IN THE ABSTRACT, SINCE IT COULD TO MUCH IMPLICIT. WE USED “STEADY” THAT SHOULD BE EASIER TO UNDERSTAND.

LN31: replace is with are.

REP: CHANGED WITH “WERE”

LN 31 – 32: Suggest moving daily to after interactions

REP: CHANGED ACCORDINGLY

LN38: Based on the methods described, I do not belief much can be inferred about the turtles’ intent. Perhaps the authors were describing the application of the data towards informing future studies of turtle sociality in the wild? I would suggest revising if so

REP: THANKS TO REVIEWER SUGGESTION. WE CHANGED THE SENTENCE TO IN ORDER TO GIVE IT THE CORRECT MEANING.

LN44-47: I would recommend integrating this into the first paragraph, as a stand along sentence usually isn’t an optimal way to start a paper. Also, the sentence is a bit of a run on and could benefit from being broken into two sentences.

REP: PARAGRAPH REMOVED

LN61: Suggest replacing “definitely put an end to” with “disproved”

REP: CHANGED ACCORDINGLY

LN64. Suggest starting a new paragraph with “According to Gardner…”

REP: THE SENTENCE WAS CHANGED

LN72: Should there be an ‘a’ in front of few? Is this sentence saying that species are able to establish them or that they are not? The second half suggests that they are, but the first half is negative in structure.

REP: THE SENTENCE WAS CHANGED ACCORDINGLY

LN74: I do not understand what “state-of-the-art” means in the context of this sentence.

REP: THE SENTENCE WAS CHANGED

LN91: Replace “a set of” with sample size.

REP: THE NUMBER OF INDIVIDUALS INVOLVED IN THE RESEARCH IS EXPLICATED IN THE NEXT PARAGRAPH

LN183: Replace visual with visually

REP: CHANGED ACCORDINGLY

LN336: Replace ‘the’ with ‘they’?

REP: THE SENTENCE WAS CHANGED

LN360: Do the authors mean elements of disturbance?

REP: CHANGED ACCORDINGLY

LN362: Replace ‘has’ with ‘have’

REP: CHANGED ACCORDINGLY

LN368: Italicize species name.

REP: CHANGED ACCORDINGLY

LN384-387: This sentence is very hard to understand. I would suggest revising

REP: THANKS TO REVIEWER SUGGESTION. WE CHANGED THE SENTENCE TO IN ORDER BE CLEARER